# *Plasmodium* and intestinal parasite infections among pregnant women at first antenatal care contact in northwest Ethiopia: A study of prevalence and associated risk factors

Zemenu Tamir [1,2]*, Abebe Animut[2], Sisay Dugassa[2], Mahlet Belachew[3], Adugna Abera [3], Aster Tsegaye[1], Berhanu Erko[2]

1 Department of Medical Laboratory Sciences, College of Health Sciences, Addis Ababa University, Addis Ababa, Ethiopia, 2 Aklilu Lemma Institute of Pathobiology, Addis Ababa University, Addis Ababa, Ethiopia, 3 Malaria and Neglected Tropical Diseases Research Team, Ethiopian Public Health Institute, Addis Ababa, Ethiopia

* zemenut266@gmail.com

## Abstract

### Background

Parasitic infections in pregnancy are detrimental for both the mother and her fetus. Malaria and intestinal parasite infections among pregnant women at their first antenatal care contact (ANC1) could offer information on their burden in pregnancy, community-level transmission, and intervention coverage, which is vital for targeted interventions. However, data is scarce in Ethiopia. This study investigated *Plasmodium* and intestinal parasite infections along with their associated risk factors among pregnant women at their ANC1 in northwest Ethiopia.

### Materials and methods

A cross-sectional study was conducted among 538 pregnant women attending ANC1 at selected health facilities in Jawi District, northwest Ethiopia, between November 2021 and July 2022. *Plasmodium* infection was diagnosed by light microscopy, rapid diagnostic tests, and multiplex real-time PCR. Whereas, intestinal parasite infections were examined microscopically using stool wet mount and Kato-Katz techniques. Predictors of *Plasmodium* and intestinal parasite infections were evaluated using logistic regression analysis. A P-value of <0.05 was used to indicate statistical significance.

### Results

Overall, 43.1% (95% CI: 38.9–47.4%) of women were infected with intestinal parasites, 19.1% (95% CI: 15.9–22.7%) had *Plasmodium* infections, and 11.7% (95% CI: 9.1–14.7%) were co-infected with *Plasmodium* and intestinal parasites. About 84.9% of malaria cases were asymptomatic, and 39.8% were sub-microscopic infections. Younger maternal age (adjusted odds ratio (AOR) = 2.01, 95% confidence interval (CI): 1.1, 3.65), primigravidity

**Data availability statement:** All relevant data are within the paper and its Supporting Information files.

**Funding:** This study was financially supported by the School of Graduate Studies, Addis Ababa University as part of the thematic research funded by the Office of Vice President for Research and Technology Transfer (VPRTT) of Addis Ababa University. The funder had no role in study design, data collection and analysis, the decision to publish, or preparation of the manuscript.

**Competing interests:** The authors declare that they have no competing interests.

(AOR = 2.37, 95% CI: 1.43, 3.92), lack of insecticide-treated bed net use (AOR = 2.58, 95% CI: 1.26, 5.3), undernutrition (AOR = 1.89, 95% CI: 1.13, 3.15), and intestinal helminth infection (AOR = 2.09, 95% CI: 1.31, 3.36) were significant predictors of *Plasmodium* infection. Whereas, rural residency (AOR = 1.62, 95% 1.03, 2.57), habit of soil eating (AOR = 2.06, 95% CI: 1.1, 3.9), consuming raw vegetables and fruits (AOR = 1.59, 95% CI: 1.09, 2.3), and lack of latrine use (AOR = 1.66, 95% CI: 1.06, 2.6) were significantly associated with intestinal parasite infections in pregnancy.

## Conclusion

High prevalences of *Plasmodium* and intestinal parasite infections were observed among pregnant women in northwest Ethiopia. These findings highlight the importance of strengthening prevention and control measures for parasitic infections and implementing malaria screening at ANC1, particularly for young and primigravid women.

## Background

Pregnancy-associated changes increase susceptibility to infections in women [1,2]. Malaria and intestinal parasite infections in pregnancy are major public health problems in tropical and subtropical countries [3,4]. In 2021, more than 13 million pregnant women in 38 malaria-endemic countries within the World Health Organization's (WHO) African Region were exposed to malaria [4]. Similarly, an estimated 24 to 70% of pregnant women were infected with intestinal parasites, and about 10% harbored more than one parasite, especially in tropical and subtropical areas with poor access to clean water, sanitation, and hygiene [3].

Ethiopia has set goals of breaking the transmission of soil-transmitted helminths and schistosomiasis by 2025 [5], and eliminating malaria by 2030 [6], and making great efforts. Despite efforts, malaria and intestinal parasite infections continue to be major public health issues in Ethiopia, especially among vulnerable populations such as pregnant women. The pooled prevalences of malaria and intestinal parasite infections among pregnant women in the country were 12.72% [7] and 27.32% [8], respectively, which were relatively higher compared to the general population.

*Plasmodium* infections cause negative impacts on a pregnant woman and her fetus [9]. *Plasmodium* infections among pregnant women in low and unstable malaria transmission areas might cause severe illness that results in severe anemia, spontaneous abortion, stillbirth, and maternal death, mainly in adolescent and primigravid women [10]. On the other hand, the infection is largely asymptomatic and sub-microscopic in high and stable transmission areas particularly among advanced aged multigravid women, which will be left untreated and be a potential source of new infections [11]. These asymptomatic and submicroscopic infections could cause adverse pregnancy outcomes such as maternal anemia and low birth weight (LBW) [12].

Similarly, intestinal parasite infections cause adverse maternal and newborn outcomes [9]. Intestinal parasite infections result in symptoms such as anorexia, abdominal pain, nausea, vomiting, diarrhea, and even intestinal bleeding. These manifestations lead to loss of appetite, malabsorption, and endogenous nutrient loss, which results in maternal anemia, undernutrition, LBW, and increased susceptibility to infections [13].

Given the significant health implications of *Plasmodium* and intestinal parasite infections in pregnancy, it is crucial to understand the underlying risk factors that exacerbate these conditions. In addition to maternal age and gravidity, socio-economic factors such as maternal

education, occupation, and income influence *Plasmodium* and intestinal parasitic infections in pregnancy. Illiterate women are less aware of the risk of malaria and intestinal parasite infections in pregnancy, as well as the prevention and treatment strategies. Similarly, women engaged in farming activities in Ethiopia are low-income and mostly reside in rural areas of Ethiopia where there is low health service accessibility and low hygiene and sanitation facilities [14,15].

To reduce the burden of intestinal parasite infections in pregnancy, the WHO recommended deworming of pregnant women after the first trimester in areas where anemia is 40% or higher in pregnancy and the burden of trichuriasis and hookworm infection is 20% and above [16]. Similarly, the usage of insecticide-treated bed nets (ITNs), effective management of malaria cases, and the administration of intermittent preventive treatment with sulphadoxine-pyrimethamine in areas with moderate to high *P. falciparum* transmission [17] are recommended to alleviate the burden of malaria in pregnancy. Ethiopia has adopted the WHO recommendations and practices ANC platform-based interventions such as deworming of pregnant women, promotes the use of ITNs, effective management of malaria cases, and provides health education on disease prevention and healthy pregnancy [18,19].

First ANC contacts attending pregnant women are suggested as an easy-to-access population segment for the surveillance of malaria [20]. It provides information on temporal trends, geographic distribution, and intervention coverage in the population, such as ITN utilization, drainage of vector breeding sites, indoor residual spray (IRS), and effective case management [21]. Such information is vital to reassess interventions and guide resource allocation, especially during the elimination phase. However, there is a paucity of data in Ethiopia on malaria and intestinal parasite infections among pregnant women before enrolment in the ANC services. Thus, this study aimed at determining the prevalence of *Plasmodium* and intestinal parasite infections and their risk factors among pregnant women in an area of high malaria and intestinal parasite transmission, Jawi District, northwest Ethiopia, prior to enrolment in the ANC platform-based interventions.

## Materials and methods

### Study area and settings

This study was conducted at selected health facilities in Jawi District, northwest Ethiopia. Jawi District, a remote hard-to-reach district in the Awi Zone of Amhara Regional State, is located about 148 kilometers southwest of Bahirdar and 576 kilometers northwest of Addis Ababa. Its mean temperature was 25.3°C (18.2–32.4°C), and the altitude ranges from 648 to 1300 meters above sea level. The district has a mean annual rainfall of 1569.4 mm. The total population of the district was 146364 in 2022, of which 76,554 were males and 69,810 were females (Jawi District Administrative Office, unpublished document). Moreover, it is among the high malaria and intestinal parasite infection transmission areas in the Amhara Regional State and a destination for seasonal migrant workers, as described elsewhere [22–24].

Bambluk Health Center, Jawi Health Center, and Jawi Primary Hospital were selected and used as study health facilities based on local malaria endemicity and accessibility. Bambluk Health Center (BHC) is a rural health center about 30 kilometers northwest of Fendika town in Bambluk Kebele. Jawi Health Center (JHC) is located in Fendika town and is the long-serving health facility in Jawi District. Jawi Primary Hospital (JPH) is the only hospital in the district located in Fendika town, which was constructed and started service delivery in 2018.

### Study design and participants

A health-facility-based cross-sectional study was conducted in Jawi District, northwest Ethiopia. This design was chosen as it allows for the rapid collection of prevalence data at a single

point in time, ideal for assessing disease burden in specific populations. Women study participants were recruited between November 27, 2021 and July 15, 2022. The inclusion criterion was pregnant women who visited the study health facilities for ANC1 in their current pregnancy during the study period. Attending ANC service for the second time or more during the current pregnancy, being HIV positive, and having a history of treatment for malaria and/or intestinal parasite infections within the past month were the exclusion criteria. Consented pregnant women who met the eligibility criteria were included in the study.

## Sample size determination and sampling technique

Sample size for prevalence of *Plasmodium* species and intestinal parasite infections was determined using a single population proportion formula, $n = \frac{(Z\alpha/2)^2 P(1-P)}{d^2}$, assuming that the prevalence (P) of *Plasmodium* infections was 8.2% [25] and intestinal parasite infections was 37.3% [26] from similar studies in Ethiopia, at a $Z_{\alpha/2} = 1.96$ at 95% confidence level and a margin of error (d) of 4%. The sample size for *Plasmodium* and intestinal parasite infections among the first ANC attending pregnant women was determined separately, and a larger sample size was taken since each pregnant woman was supposed to be diagnosed for both *Plasmodium* and intestinal parasite infections. The larger minimum calculated sample size was obtained for intestinal parasite infections, which was 532 women.

For determination of the risk factors of *Plasmodium* and intestinal parasite infections, the sample size was determined by a double proportion formula using Epi Info version 7.2.2.6, considering different risk factors identified in the literature such as maternal age, gravidity, maternal education, occupation, residence area, undernutrition, ITN use, IRS status, larine utilization, family water source, shoe-wearing practice, and geophagia. Among the predictors, the largest calculated minimum sample size was obtained for gravidity, assuming a 16.1% prevalence of *Plasmodium* infections among primigravidae and a 6.5% prevalence among multigravidae from a study in northwest Ethiopia [27], at a 95% confidence level, 80% power, and an equal number of primigravidae and multigravidae. The larger minimum calculated sample size was obtained for *Plasmodium* infections, which was 380 women. Hence, 538 first ANC1 attending eligible pregnant women were recruited and enrolled.

To determine the number of participants required from each health facility, participants were assigned using a probability proportionate to size sampling technique. The number of pregnant women from each health facility was determined based on the proportion of pregnant women who attended ANC1 at each health facility in the last 6 months prior to the study (Fig 1). Thus, 252 ANC1-attending pregnant women were enrolled from JPH, 185 from JHC, and 101 from BHC.

Participants were selected using a systematic random sampling technique. Based on the estimated average number of daily visitors for ANC1 (N) and the number of women to be recruited per day in each health facility (n), the interval (k) was calculated as $k = N/n$, and the first participant was determined by the lottery method from those coming 1 to k.

## Data collection procedure

After ANC1 attending women provided consent to participate in the study, data was collected using a structured and pretested questionnaire and checklists. The questionnaire and checklists were initially prepared in English and later translated to the local languages (Amharic and Awungi) by native speakers of the languages. To ensure consistency, data collectors read and completed the Amharic or the Awugni questionnaire for the participants, regardless of the participants' literacy level [23].

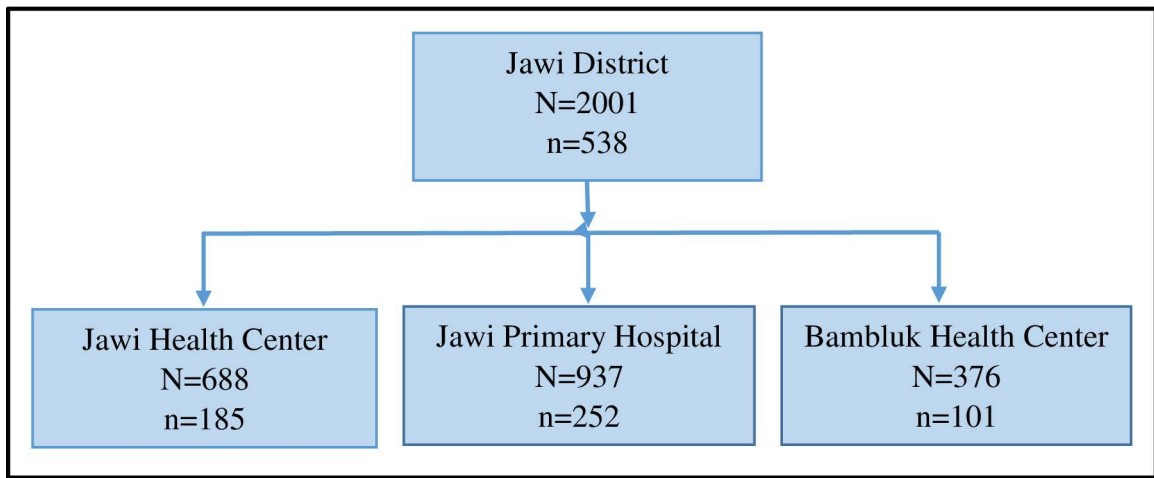

**Fig 1. Sampling procedure of study participants from study health facilities.** N = number of first ANC contact attending women in the last 6 months in each health facility, n = number of first ANC attending pregnant women enrolled in each health facility.

The questionnaire included the participants socio-demographic characteristics, malaria prevention practices (ITN ownership and usage and IRS status), drinking water source, habit of soil eating (geophagia) during the current pregnancy, availability of latrine and usage, practice of eating raw vegetables and fruit, shoe wearing, and hand washing practice after toilet. Information on the obstetric and clinical profiles of women, such as gravidity, parity, status of malaria and/or intestinal parasite infection symptoms, and usage of anti-helminthic and anti-malarial drugs, was collected by interview and from follow-up cards.

In the health centers, gestational age was determined by the last menstrual period, whereas the Mindray DP-50 digital ultrasound machine was used in the hospital. A digital thermometer was used to take the mother's body temperature under her arms. Mid-upper arm circumference (MUAC) was measured to the nearest 0.1 centimeters using an insertion-type MUAC tape that is non-elastic and non-stretchable, as described in the previous work [23]. Undernutrition was defined as MUAC less than 23 centimeters [28].

### Laboratory analysis

The laboratory analysis and data collection involved two steps: first, stool samples were collected and examined microscopically using the wet mount and Kato-Katz techniques for intestinal parasites. Blood samples were then collected for *Plasmodium* infection diagnosis via light microscopy, rapid diagnostic tests (RDT), and real-time PCR. While RDT and microscopy are widely used for malaria diagnosis, real-time PCR was employed to detect sub-microscopic infections, ensuring a more accurate estimation of *Plasmodium* prevalence.

**Intestinal parasite diagnosis.** Pregnant women were given clean plastic stool cups with wooden applicator sticks and instructed to bring about 2 grams of their own stool. A wet-mount smear and two Kato-Katz (KK) slides (using a 41.7 mg template) were prepared from a single stool specimen of each woman. The wet mount slides were examined microscopically within 30 minutes of collection for detection of trophozoites and/or cysts of intestinal protozoa and larvae or ova of intestinal helminths. Similarly, the KK preparations were examined within one hour for the detection and quantification of hookworm ova and within 24 hours for the other intestinal parasites [29].

Stool specimen collection, transportation, processing, storage, examination, and other laboratory procedures were conducted following the recommended standard operational procedures [29]. A Stool specimen was collected, processed, and examined by two trained medical laboratory professionals working in each health institution, cross-checked by a third more experienced professional.

A stool specimen was considered positive for intestinal parasite*s* if trophozoites and/or cysts of intestinal protozoa and/or larvae or eggs of intestinal helminths were detected by either of the two or both methods. A specimen was considered negative for intestinal parasites if trophozoites, cysts, larvae, and eggs of parasites were not detected from the wet mount smear and two KK-thick smear slides [29].

**_Plasmodium_ infection diagnosis using light microscopy and rapid diagnostic tests.** At the health facility level, *Plasmodium* infection was diagnosed using RDT and light microscopy. The RDT testing was performed from the finger prick capillary blood using the Abbott Bioline™ Malaria Ag P.f/P.v test kit (Standard Diagnostics, Inc., Suwon, Republic of Korea) as per the manufacturer's instructions. The test detects Histidine-Rich Protein II (HRP-II) specific to *P. falciparum* and *Plasmodium* lactate dehydrogenase (pLDH) specific to *P. vivax*.

Capillary blood specimen collection, preparation of blood smears, and staining procedures were previously described [23]. Thick and thin blood smears were examined by two independent, experienced laboratory technologists who were blinded to the RDT results and cross-checked by a third, more experienced laboratory technologist in case of discrepancies. A slide was classified as negative if no asexual forms or gametocytes of *Plasmodium* species were detected after examining at least 200 high-power microscope fields [30]. The asexual parasitemia level of *Plasmodium* infections was estimated among women who had microscopic *Plasmodium* parasitemia. The asexual parasite density of *P. falciparum*, *P. vivax*, and *P. falciparum* and *P. vivax* mixed infections was estimated on the thick film against 200 leucocytes, assuming a total white blood cell count of 8000/μl [23].

**Detection of _Plasmodium_ parasites using real-time polymerase chain reaction.** Dry blood spots (DBSs) were collected, and qPCR testing was performed for 420 pregnant women; 78 women were diagnosed positive for *Plasmodium* parasites by microscopy and/or RDT, and 342 women were negative by both microscopy and RDT. Dry blood spot collection, preparation, transportation, and storage are detailed elsewhere [23]. The polymerase chain reaction assay was done at the Ethiopian Public Health Institute National Parasitology Laboratory.

Genomic DNA (gDNA) extraction was performed using the Geneius™ Micro gDNA Extraction Kit (Geneaid Biotech Ltd., Taipei, Taiwan) as detailed in our previous work [23]. All microscopy and/or RDT-positive samples were extracted and analyzed individually. Whereas *Plasmodium* infection prevalence estimation for microscopy and RDT negative samples was performed using pooled DBS sample extraction and analysis with slight modifications of Zhou et al. [31] and detailed elsewhere [23] (Fig 2). Briefly, to achieve adequate lysis, ten punched-out circles with a 3-mm diameter were pooled together in 2ml Eppendorf tubes and incubated overnight with lyse buffer and proteinase K solution. Subsequently, DNA extraction was carried out in accordance with standard protocols. The isolated DNA was tested using multiplex real-time genus PCR. For genus qPCR, samples in negative pools were considered negative. For pools with a positive genus qPCR result, individual DBSs were extracted in accordance with the protocol and genus-specific qPCR, and then species-specific qPCR assays were carried out [23].

The QuantStudio 5 real-time PCR system (Applied Biosystems) was used to carry out TaqMan fluorescence-based DNA amplification and detection. The multiplex real-time PCR assay was conducted in a final volume of 10μl in two cycles. All samples were tested by

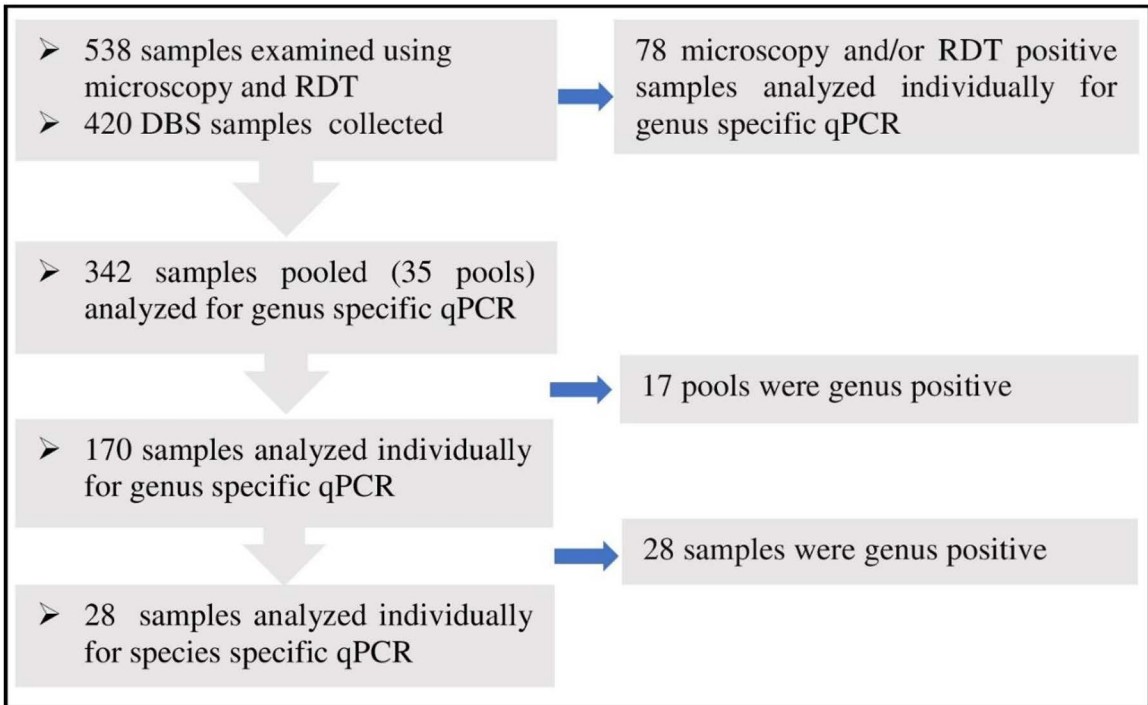

**Fig. 2. A flow chart showing dry blood spot pooling and analysis in real-time polymerase chain reaction.**

multiplexing pan-*Plasmodium*-specific and *P. falciparum*-specific primers in the first round, whereas *P. falciparum* and *P. vivax*-specific primers were multiplexed during the second round, as described in Belachew *et al.* and Tamir *et al.* [23,32] (S1 Table).

The following thermal cycling conditions were used during PCR amplifications: 95 °C for 1 minute, followed by 45 cycles of 95 °C for 15 seconds and 57 °C for 45 seconds for the first PCR run; and 95 °C for 1 minute, followed by 45 cycles of 95 °C for 15 seconds and 53 °C for 45 seconds for the second run. The 3D7 DNA standard was run as a positive control and nuclease-free water as a negative control in each test. For the PCR run, all samples that were considered qualified runs had Ct values of < 30.0 for HsRNaseP and Ct values between 25.0 and 30.0 for the positive control. Positive samples had Ct values between 12 and 40 and a sigmoidal-shape amplification curve [23,32].

## Operational definitions

Asymptomatic *Plasmodium* infection: a positive RDT and/or detection of *Plasmodium* parasites using light microscopy and/or real-time qPCR in the absence of malaria symptoms, a history of fever within the past 48 hours, and axillary temperature < 37.5 °C.

Symptomatic *Plasmodium* infection: a positive RDT and/or detection of *Plasmodium* parasites using light microscopy and/or real-time qPCR from women who exhibited at least one of the signs and/or symptoms of malaria, like axillary temperature ≥ 37.5 °C, fever, joint pain, malaise, vomiting, chills, etc., during ANC contact or within the past 48 hours.

Sub-microscopic *Plasmodium* infection or sub-microscopic malaria: having a positive RDT and/or detection of *Plasmodium* parasites by real-time qPCR but not by microscopy.

Subpatent infection: *Plasmodium* parasites detected by qPCR but not detected by microscopy and negative by malarial RDT.

Multigravida: A woman who is pregnant for the second time and above.
Timely ANC initiation: ANC1 is initiated in the first trimester of pregnancy.
Delayed ANC initiation: ANC1 is initiated after the first trimester of pregnancy.

## Statistical analysis

The collected data included demographic, environmental, and behavioral variables, which were used in subsequent statistical analyses. The data was cleaned and double entered into EpiData 3.1 software and then exported to Statistical Package for Social Sciences (SPSS) version 25 statistical software (IBM Corp., New York, USA) and analyzed. The prevalence of intestinal parasites and *Plasmodium* species infections was estimated by dividing the number of pregnant women diagnosed positive by the total number of pregnant women examined.

A chi-square test was used to assess the relationship among prevalences of intestinal parasites and *Plasmodium* species infections with independent variables. Bivariable and multivariable logistic regression models were used to identify factors that contributed to intestinal parasite and *Plasmodium* infections. Only variables with P-values < 0.05 in the bivariable logistic regression analysis and those indicated to be associated with *Plasmodium* and intestinal parasite infections in previous studies were included in the multivariable logistic regression model. The odds ratio (OR) with a 95% confidence interval (CI) was used to measure the strength of the statistical association. A P-value < 0.05 was used to indicate statistical significance.

## Ethical consideration

This study was conducted in accordance with the Declaration of Helsinki. The study protocol was approved by the Institutional Review Board (IRB) of Aklilu Lemma Institute of Pathobiology, Addis Ababa University (reference number: ALIPB IRB/60/2013/21). Permission to conduct the study was obtained from the Amhara Public Health Institute, the Awi Zone Health Office, the Jawi District Health Office, and the management of each health facility after having thorough discussions on the procedures and purpose of the study. Written informed consent was obtained from pregnant women after they were briefed about the goal and advantages of the study using their mother tongue language and level of comprehension. They were also made aware of their complete freedom to discontinue participation in the study at any moment. To protect the privacy of the participant women and confidentiality of the data, all personal identifiers were removed, only codes were utilized, and the data was stored securely.

Pregnant women got free clinical and laboratory diagnoses, and copies of the results were delivered to the attending clinicians. The infected women were treated as per the national treatment guidelines[33].

## Results

A total of 538 first-ANC contact-attending pregnant women were involved in the study. The median (interquartile range) age of the women was 25 (8) years, and the majority (78.6%) were young adults aged 20–34 years (Table 1). About 94.1% were married, 59.1% were rural residents, 44.2% were engaged in farming, and 70.8% lived in households with less than five members. The average gestational age (standard deviation (SD)) at their first ANC contact was 20.29 (6.57) weeks (5.9–36.7 weeks). About 81.8% (95% CI: 78.3–85.0) of the women attended their first ANC contact delayed in their second and/or third trimesters of pregnancy. More than 56% of the women were primigravidae or secundigravidae. Among the women, 22.5% were undernourished, 6.5% had signs and symptoms of malaria, and 10.6% reported symptoms of intestinal infections.

**Table 1. Socio-demographic, obstetrical, and clinical profiles of study participant pregnant women, northwest Ethiopia, 2022.**

| Characteristics | Category | n (%) |
|---|---|---|
| Age category in years | ʻ20 | 75(13.9) |
| | 20-34 | 423(78.6) |
| | ≥35 | 40(7.4) |
| Residence | Urban | 220(40.9) |
| | Rural | 318(59.1) |
| Marital status | Single | 32(5.9) |
| | Married | 506(94.1) |
| Education status | Illiterate | 244(45.4) |
| | Primary education | 169(31.4) |
| | Secondary education | 74(13.8) |
| | College and above | 51(9.5) |
| Occupation | House wife | 174(32.3) |
| | Farmer | 238(44.2) |
| | Trader | 79(6.9) |
| | Civil servant | 37(14.7) |
| | Others | 10(1.9) |
| Family size | <5 | 381(70.8) |
| | ≥5 | 157(29.2) |
| Gravidity | Primigravida | 163(30.3) |
| | Secundigravida | 139(25.8) |
| | Multigravida | 236(43.9) |
| Gestational Age | First trimester | 98(18.2) |
| | Second trimester | 356(66.2) |
| | Third trimester | 84(15.6) |
| ANC initiation | Timely | 98(18.2) |
| | Delayed | 440 (81.8) |
| Undernutrition | No | 417(77.5) |
| | Yes | 121(22.5) |
| Malaria signs and symptoms | Yes | 35 (6.5) |
| | No | 503(93.5) |
| Intestinal parasite infection symptoms | Yes | 57(10.6) |
| | No | 481(89.4) |

Among the enrolled pregnant women, 68.2% reported that they had at least one ITN (Table 2). However, about 63% of them did not consistently sleep under it. More than 65% reported that their house was not sprayed with IRS within the last 12 months. About 55% wore shoes, 66.9% owned and consistently used latrine, 68.4% washed their hands after toilet, and 43.1% used protected tap water for drinking. About 9.5% practiced soil eating (geophagia), and 45% had frequent river water contact during the current pregnancy.

### *Plasmodium* and intestinal parasite infections at first antenatal care contact

In total, 50.6% (95% CI: 46.2–54.9) of the pregnant women had at least a single species of *Plasmodium* or intestinal parasite infections. The overall *Plasmodium* infection was 19.1% (95% CI: 15.9–22.7%) (Table 3). *Plasmodium falciparum* infected 15.8% of the women, and *P. vivax* infected 2.6%. Four pregnant women (0.7%) had *P. falciparum* and *P. vivax* mixed

**Table 2. Malaria and intestinal parasite infection prevention practices of pregnant women, Jawi District, northwest Ethiopia, 2022.**

| Variables | Response | n (%) |
|---|---|---|
| Ownership of ITN | No | 171(31.8) |
| | Yes | 367(68.2) |
| Consistent use of ITNs (n = 367) | No | 232(63.2) |
| | Yes | 135(36.8) |
| IRS in the last 12 months | No | 352(65.4) |
| | Yes | 186(34.6) |
| Eating soil (Geophagia) | No | 487(90.5) |
| | Yes | 51(9.5) |
| Routine shoe wearing practice | Yes | 295(54.8) |
| | No | 243(45.2) |
| Availability and consistent use of latrine | Yes | 360(66.9) |
| | No | 178(33.1) |
| Routine hand washing practice after toilet use | Yes | 368(68.4) |
| | No | 170(31.6) |
| Drinking water source | Tap | 232(43.1) |
| | Well | 147(27.3) |
| | Spring | 149(29.6) |
| River water contact | No | 294(54.6) |
| | Yes | 244(45.4) |

**Table 3. Prevalence of *Plasmodium* species infections among pregnant women at their first ANC contact, northwest Ethiopia.**

| *Plasmodium* infections | n (%) |
|---|---|
| *Plasmodium* species infections | 103(19.1) |
| *Plasmodium falciparum* infections | 85(15.8) |
| *Plasmodium vivax* infection prevalence | 14(2.6) |
| Mixed *P.falciparum* and *P.vivax* infections | 4(0.7) |
| Asymptomatic *Plasmodium* infection[a] | 87(17.3) |
| Symptomatic *Plasmodium* infection[b] | 16(45.7) |
| Microscopic *Plasmodium* infection prevalence | 62(11.5) |
| Sub-microscopic *Plasmodium* infection prevalence | 41(7.6) |
| Subpatent *Plasmodium* infection prevalence[c] | 25(7.3) |

[a]calculated from asymptomatic pregnant women, n = 503; [b] Calculated form symptomatic pregnant women, n = 35;calculated from microscopy and RDT negative women which were diagnosed by qPCR (n = 342)

infections. *Plasmodium falciparum* accounted for 82.5% (85/103) of the identified *Plasmodium* infections (Fig 3a). The prevalence of asymptomatic *Plasmodium* infection was 17.3% (87/503), which accounted for 84.5% of the infected cases (Fig 3b). Similarly, the prevalence of microscopic, sub-microscopic, and subpatent *Plasmodium* infections was 11.5%, 7.6%, and 7.3%, respectively. Among the *Plasmodium* infections, 60.2% were microscopic, 39.8% were sub-microscopic, and 24.3% were subpatent (Fig 3c). The mean (SD) asexual malaria parasitemia among microscopic cases was 9829.2 (16152.7) parasites/microliter of blood. Similarly, gametocytes were detected among nine women with a mean (SD) gametocyte density of 1475.6 (1219.6) parasites/microliter of blood.

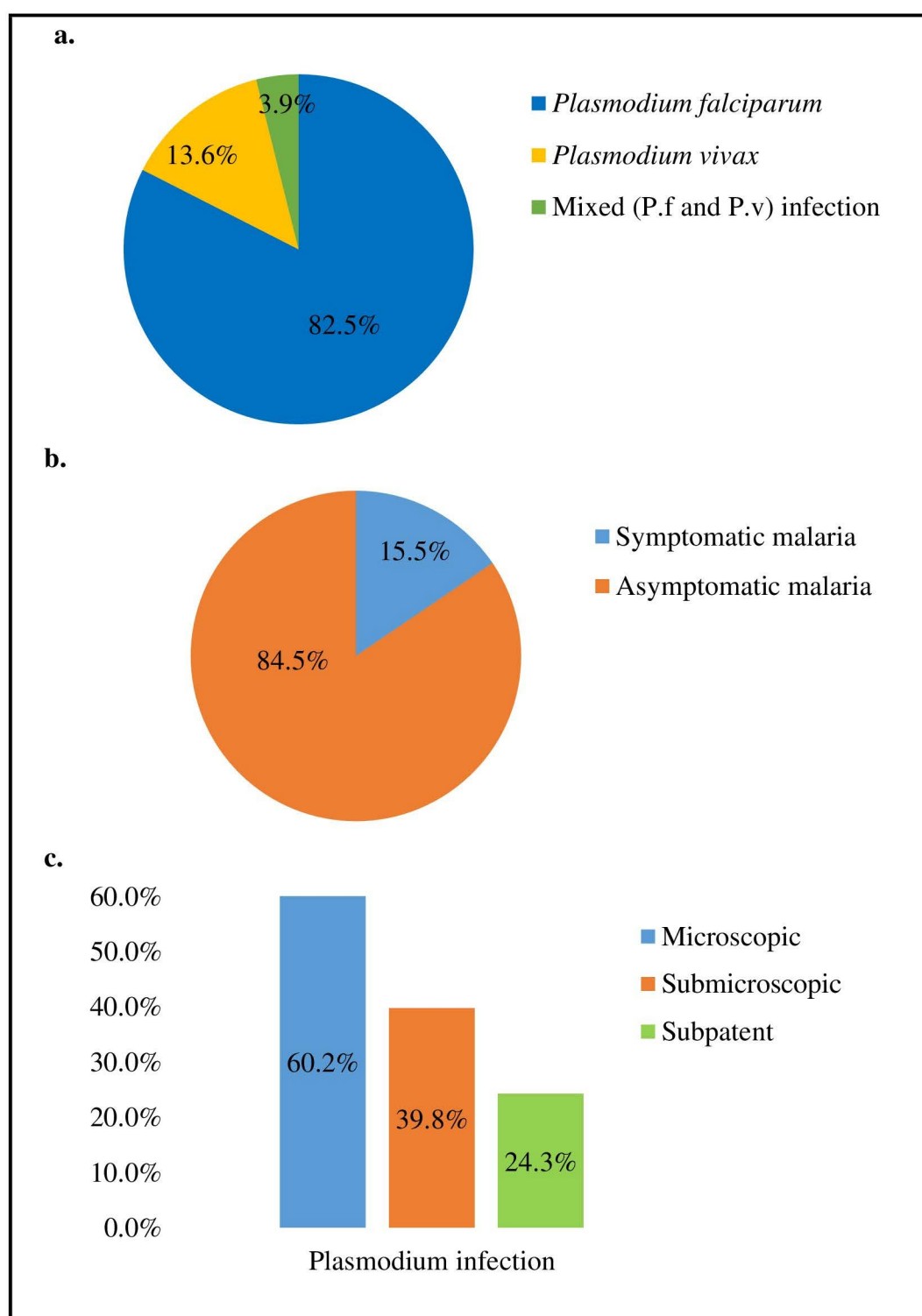

**Fig 3. Shows the distribution of *Plasmodium* species, the proportion of symptomatic and asymptomatic *Plasmodium* infections, and the proportion of microscopic, submicroscopic, and subpatent *Plasmodium* infections among pregnant women at first ANC contact, northwest Ethiopia.**

Two hundred thirty-two pregnant women (43.1%, 95% CI: 38.9–47.4) were infected with at least a single species of intestinal parasite (Table 4). Of these, 72.4% (168/232) harbored a single infection, 23.3% (54/232) co-infections, and 4.3% (10/232) triple infections. Intestinal helminths infected 36.6% (197/538) of the participants, which accounted for 84.9% of the intestinal parasite infections, whereas intestinal protozoa infected 13.8% (74/538) of the enrolled pregnant women. Nine species of intestinal parasites were identified. Among these, hookworm was the most frequent intestinal parasite (20.6%, n = 111; manifested as 62 single and 49 double or triple infections), followed by *S. mansoni* (12.5%, n = 67; manifested as 36 single and 31 double or triple infections) and *G. lamblia* (8%, n = 43; manifested as 18 single and 25 double or triple infections).

*Plasmodium* and intestinal parasite coinfections were found in 11.7% of the pregnant women (Fig 4). Coinfections of *Plasmodium* with intestinal helminths, intestinal protozoa, and both were observed in 8.7%, 1.1%, and 1.9% of the women, respectively.

## Factors associated with *Plasmodium* and intestinal parasite infections in pregnancy

This study showed that adolescent (age < 20 years) pregnant women had a 2.01 times increased risk of *Plasmodium* infection compared to adult (age ≥ 20 years) women (AOR = 2.01, 95% CI: 1.1, 3.65) (Table 5). Primigravid women were 2.37 times more likely to acquire *Plasmodium* infection than their multigravid counterparts (AOR = 2.37, 95% CI: 1.43, 3.92). *Plasmodium* infections showed a significant association with a lack of practice of malaria prevention activities. Pregnant women who did not use ITN consistently and those who lived in households that did not spray in the last year had about 2.58 times and 1.77 times increased likelihood of

**Table 4. Prevalence of intestinal parasite infections among pregnant women at first ANC contact, northwest Ethiopia, 2022.**

| Parasite species detected | n (%) |
|---|---|
| Hookworm | 62(11.5) |
| *S. mansoni* | 36(6.7) |
| *G. lamblia* | 18(3.3) |
| *H. nana* | 21(3.9) |
| *E. histolytica* | 15(3.0) |
| *A. lumbricoides* | 6(1.1) |
| *S. stercoralis* | 4(0.7) |
| *Tania species* | 4(0.7) |
| *T. trichiura* | 2(0.4) |
| Hookworm co-infection or triple infection with other intestinal parasites | 49(9.1) |
| *S.mansoni* co-infection or triple infection with other intestinal parasites | 31(5.8) |
| *G.lamblia* co-infection or triple infection with other intestinal parasites | 25(4.6) |
| Total intestinal parasite co-infection | 54(10) |
| Total intestinal parasite triple infection | 10(1.86) |
| Total intestinal helminth infection | 197(36.6) |
| Intestinal helminth infection without protozoa | 158(29.4) |
| Total intestinal protozoa infection | 74(13.8) |
| Intestinal protozoa infection without helminth | 35(6.5) |
| Intestinal helminth and intestinal protozoa multiple infection | 39(7.2) |
| Overall intestinal parasitic infection | 232(43.1) |

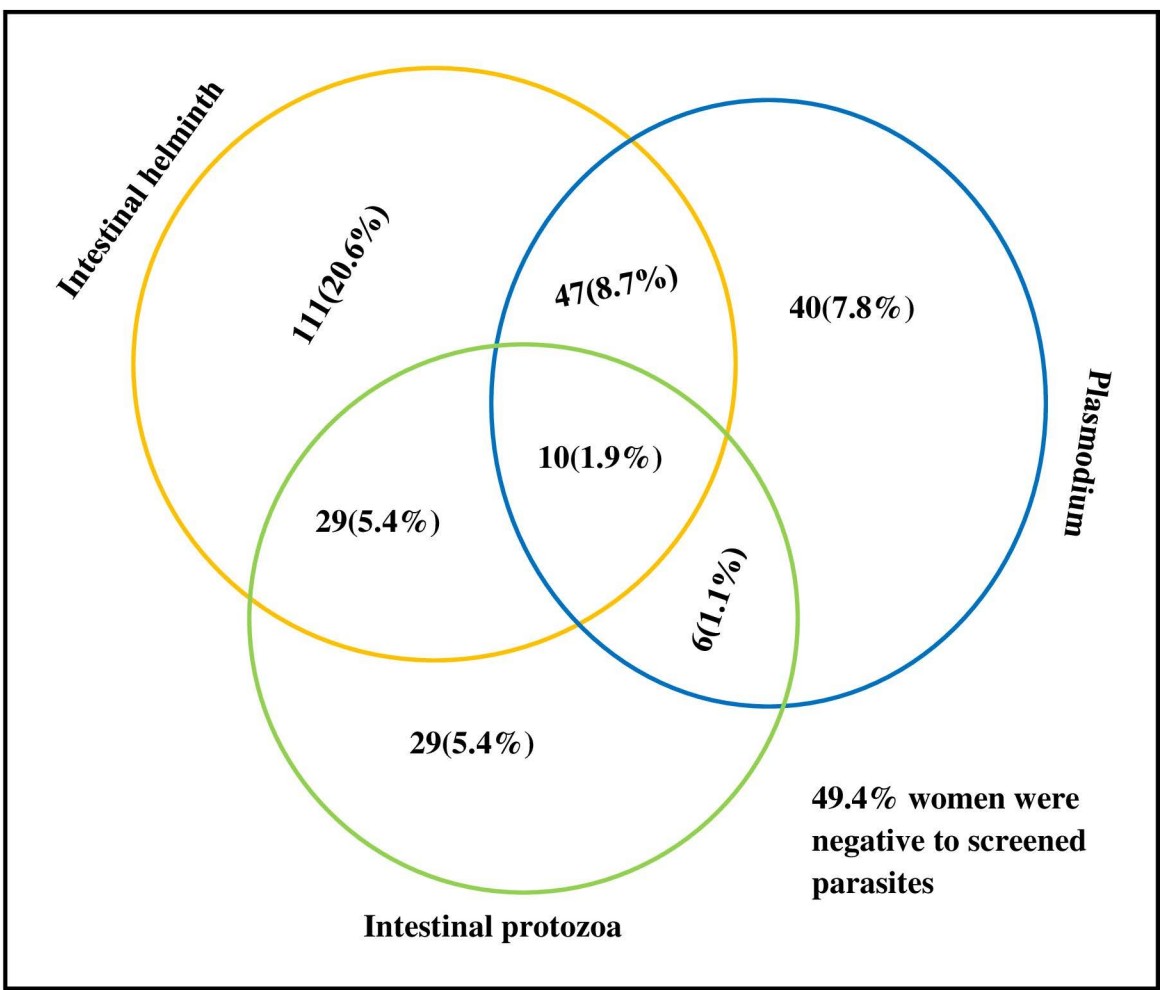

**Fig. 4. Venn diagram displaying the overlap of Plasmodium species, intestinal helminth, and intestinal protozoa infections among pregnant women at first ANC contact, northwest Ethiopia, 2022.**

*Plasmodium* infection compared to those who slept under ITN regularly (AOR = 2.58, 95% CI: 1.26, 5.3) and lived in households sprayed with IRS (AOR = 1.77, 95% CI: 1.03, 3.05), respectively. Moreover, undernourished and intestinal-helminth-infected pregnant women had about a double increased likelihood of *Plasmodium* infection compared to those women who were well-nourished (AOR = 1.89, 95% CI: 1.13, 3.15) and intestinal-helminth-uninfected (AOR = 2.09, 95% CI: 1.31, 3.36) during first ANC contact.

On the other hand, this study showed that pregnant women living in rural villages were 1.62 times more likely to be infected with intestinal parasites than their urban counterparts (AOR = 1.62, 95% 1.03, 2.57) (Table 6). Intestinal parasite infections were significantly associated with the habit of eating soil during pregnancy, consuming raw or unwashed vegetables and fruits, and the quality of potable water in the household. Women who had the habit of eating soil in pregnancy had more than a twofold higher risk of intestinal parasite infections than those who did not eat soil (AOR = 2.06, 95% CI: 1.1, 3.9), and women who had the habit of consuming uncooked vegetables/fruits had about 1.59 times increased likelihood of acquiring intestinal parasite infections (AOR = 1.59, 95% CI: 1.09, 2.3). Similarly, women who used water from unprotected sources had about 1.81 times increased risk of intestinal parasite infection than those who

**Table 5. Factors associated with *Plasmodium* infection among pregnant women at first ANC contact, northwest Ethiopia, 2022.**

| Variables | | *Plasmodium* infection, n/N* (%) | Bivariable analysis | | Multivariable analysis | |
|---|---|---|---|---|---|---|
| | | | COR@ [95% CI] | P-value | AOR$ [95% CI] | P-value |
| Age | ≥20 years | 74/463(16.0) | 1 | | 1 | |
| | <20 years | 29/75(38.7) | 3.31[1.96,5.61] | <0.001 | 2.01[1.1,3.65] | 0.024 |
| Educational background | Literate | 52/294(17.7) | 1 | | | |
| | Illiterate | 51/244(20.9) | 1.23[0.8,1.89] | 0.346 | | |
| Gravidity | Multigravida | 49/375(13.1) | 1 | | 1 | |
| | Primigravida | 54/163(33.1) | 3.29[2.12,5.14] | ˹0.001 | 2.37[1.43,3.92] | 0.001 |
| Residence | Urban | 30/220(13.6) | 1 | | 1 | |
| | Rural | 73/318(23.0) | 1.89(1.19,3.01) | 0.007 | 1.46[0.87,2.45] | 0.151 |
| Gestational age | 1st trimester | 13/98(13.3) | 1 | | 1 | |
| | 2nd trimester | 69/356(19.4) | 1.57[0.83,2.98] | 0.166 | 1.69[0.85,3.37] | 0.132 |
| | 3rd trimester | 21/84(25.0) | 2.18[1.02,4.68] | 0.046 | 1.94[0.84,4.44] | 0.119 |
| Marital status | Married | 94/506(18.6) | 1 | | | |
| | Single | 9/32(28.1) | 1.72[0.77,3.83] | 0.188 | | |
| Family size | <5 | 71/381(18.6) | 1 | | | |
| | ≥5 | 32/157(20.4) | 1.12[0.7,1.78] | 0.64 | | |
| Stagnant water | No | 92/481(19.1) | 1 | | | |
| | Yes | 11/57(19.3) | 1.01[0.5,2.03] | 0.97 | | |
| ITN use | Yes | 10/135(7.4) | 1 | | 1 | |
| | No | 93/403(23.1) | 3.75[1.89,7.44] | <0.001 | 2.58[1.26,5.3] | 0.01 |
| IRS in the last 12 months | Yes | 23/186(12.4) | 1 | | 1 | |
| | No | 80/352(22.7) | 2.08[1.26,3.45] | 0.004 | 1.77[1.03,3.05] | 0.04 |
| Protozoa infection | No | 87/464(18.8) | 1 | | | |
| | Yes | 16/74(21.6) | 1.19[0.66,2.18] | 0.56 | | |
| Undernutrition | No | 64/417(15.3) | 1 | | 1 | |
| | Yes | 39/121(32.2) | 2.62[1.65,4.18] | <0.001 | 1.89[1.13,3.15] | 0.015 |
| Helminth infection | No | 46/341(13.5) | 1 | | 1 | |
| | Yes | 57/197(28.9) | 2.61[1.69,4.04] | <0.001 | 1.97[1.22,3.18] | 0.005 |

*n = Number of *Plasmodium* infected women in that category, N = total number of women in that category, @COR = Crude Odds Ratio, $AOR=adjusted odds ratio.

used water from protected sources (AOR = 1.81, 95% CI: 1.2, 2.76). Moreover, pregnant women who did not own and use latrine consistently had a 1.66 times increased risk of intestinal parasite infections than those who used latrine routinely (AOR = 1.66, 95% CI: 1.06, 2.6).

In this study, rural pregnant women were 2.05 times more likely to be co-infected with *Plasmodium* and intestinal parasites than urban pregnant women (AOR = 2.05, 95% CI: 1.1, 3.81) (Table 7). Similarly, primigravid women had a 2.75 times increased risk of acquiring both *Plasmodium* and intestinal parasites compared to their multigravida counterparts (AOR = 2.75, 95% CI: 1.53, 4.97). Moreover, undernourished pregnant women had more than a double risk of acquiring *Plasmodium* and intestinal parasite co-infections than well-nourished women (AOR = 2.45, 95% CI: 1.39, 4.33).

## Discussion

Malaria and intestinal parasite infections in pregnancy contribute to the morbidity and mortality of women as well as adverse birth outcomes. Thus, monitoring the epidemiology of *Plasmodium* infection and intestinal parasite infections at ANC1 in high-transmission and

**Table 6. Factors associated with intestinal parasite infections among pregnant women at first ANC contact, northwest Ethiopia.**

| Variables | | IP[#] infection, n/N[*] (%) | Bivariable analysis | | Multivariable analysis | |
|---|---|---|---|---|---|---|
| | | | COR[95% CI] | P-value | AOR [95% CI] | P-value |
| Age | ≥20 years | 198/463(42.8) | 1 | | | |
| | <20 years | 34/75(45.3) | 1.11[0.68,1.81] | 0.667 | | |
| Educational status | Literate | 119/294(40.5) | 1 | | | |
| | Illiterate | 113/244(46.3) | 1.27[0.9, 1.79] | 0.174 | | |
| Occupation | Non-farmer | 125/300(41.7) | 1 | | | |
| | Farmer | 107/238(45.0) | 1.14[0.81,1.61] | 0.444 | | |
| Residence | Urban | 68/220(30.9) | 1 | | 1 | |
| | Rural | 164/318(51.6) | 2.38[1.66, 3.41] | <0.001 | 1.62[1.03,2.57] | 0.038 |
| Marital status | Married | 214/506(42.3) | 1 | | | |
| | Single | 18/32(56.3) | 1.75[0.85, 3.61] | 0.126 | | |
| Gravidity | Multigravida | 151/375(40.3) | 1 | | 1 | 0.093 |
| | Primigravida | 81/163(49.7) | 1.46[1.01, 2.12] | 0.043 | 1.47[0.98,2.2] | 0.059 |
| Gestational age | 1st trimester | 39/98[39.8] | 1 | | | |
| | 2nd trimester | 157/356(44.1) | 1.19[0.76, 1.88] | 0.446 | | |
| | 3rd trimester | 36/84(42.9) | 1.14[0.63, 2.05] | 0.676 | | |
| Undernutrition | No | 165/417(39.6) | 1 | | 1 | |
| | Yes | 67/121(55.4) | 1.89[1.26, 2.85] | 0.002 | 1.49[0.96,2.3] | 0.074 |
| Family size | <5 | 159/381(41.7) | 1 | | | |
| | ≥5 | 73/157(46.5) | 1.21[0.84, 1.76] | 0.311 | | |
| Geophagia | No | 200/487(41.1) | 1 | | 1 | |
| | Yes | 32/51(62.7) | 2.42[1.33, 4.39] | 0.004 | 2.06[1.1,3.9] | 0.026 |
| Routine shoe wearing practice | Yes | 114/295(38.6) | 1 | | 1 | |
| | No | 118/243(48.6) | 1.5[1.06, 2.11] | 0.021 | 1.25[0.82, 1.9] | 0.304 |
| Consistent use of latrine | Yes | 130/360(36.1) | 1 | | 1 | |
| | No | 102/178(57.3) | 2.37[1.65, 3.43] | <0.001 | 1.66[1.06, 2.6] | 0.026 |
| Routine handwashing after toilet use | Yes | 151/368(41.0) | 1 | | | |
| | No | 81/170(47.6) | 1.31[0.91, 1.88] | 0.15 | | |
| Family water source | Protected | 72/232(31.0) | 1 | | 1 | |
| | Unprotected | 160/306(53.2) | 2.44[1.7, 3.48] | <0.001 | 1.81[1.2, 2.76) | 0.006 |
| Eating raw/unwashed vegetables/fruits | No | 139/353(39.4) | 1 | | 1 | |
| | Yes | 93/185(50.3) | 1.53[1.07, 2.19] | 0.02 | 1.59[1.09, 2.3] | 0.017 |

[*]n = Number of intestinal parasite infected women in that category, N = total number of women in that category, [#]IP=Intestinal parasite.

hard-to-reach areas like Jawi District is important to determine the burden, reassess the control measures, and plan targeted interventions.

In the current study, 80% of women delayed initiating ANC. This high rate of delayed ANC initiation aligns with the National Demographic Health Survey [34]. WHO recommends early ANC initiation for optimal health promotion, disease prevention, and curative services [16]. In Ethiopia, particular interventions aimed at healthy pregnancy, such as deworming, the provision of ITN, and supplements (iron and folic acid), are provided on the ANC platform [19]. Thus, due to the delayed ANC1, the majority of pregnant women in the study area may not fully benefit from the services provided on the platform.

Pregnant women are more susceptible to infections than non-pregnant women [2]. Moreover, the prevalence of these infections at ANC1 is a reflection of the transmission level and

**Table 7. Factors associated with *Plasmodium*-intestinal parasite co-infections at first ANC contact of women, northwest Ethiopia, 2022.**

| Variables | | *Plasmodium*-IP co-infection, n/N*(%) | Bivariable analysis | | Multivariable analysis | |
|---|---|---|---|---|---|---|
| | | | COR[95%CI] | P-value | AOR [95% CI] | P-value |
| Age | ≥20 years | 46/463(9.9) | 1 | 0.002 | 1 | |
| | <20 years | 17/75(22.7) | 2.66[1.43,4.94] | | 1.65[0.83,3.29] | 0.156 |
| Educational status | Literate | 31/294(10.5) | 1 | | | |
| | Illiterate | 32/244(13.1) | 1.28[0.76,2.17] | 0.357 | | |
| Occupation | Non-farmer | 32/300(10.7) | 1 | | | |
| | Farmer | 31/238(13.0) | 1.25[0.74,2.12] | 0.399 | | |
| Residence | Urban | 16/220(7.3) | 1 | | 1 | |
| | Rural | 47/318(14.8) | 2.21[1.22,4.01] | 0.009 | 2.05[1.1,3.81] | 0.023 |
| Gravidity | Multigravid | 28/375(7.5) | 1 | | 1 | |
| | Primigravid | 35/163(21.5) | 3.39[1.98,5.79] | 0.001 | 2.75[1.53,4.97] | 0.001 |
| Gestational age | 1st trimester | 9/98[9.2] | 1 | | | |
| | 2nd trimester | 43/256(12.1) | 1.36[0.64,2.89] | 0.427 | | |
| | 3rd trimester | 11/84(13.1) | 1.49[0.59,3.79] | 0.402 | | |
| Undernutrition | No | 27/417(8.6) | 1 | | 1 | |
| | Yes | 36/121(22.3) | 3.04[1.76,5.26] | 0.001 | 2.45[1.39,4.33] | 0.002 |

*n=Number of *Plasmodium*-intestinal parasite co-infected women in that category, N=total number of women in that category.

intervention coverage in the community [20]. In the current study, over half of the pregnant women were infected with *Plasmodium* and/or intestinal parasites at ANC1, revealing a high burden of parasites in the study area, consistent with previous reports [22,24]. This finding exceeds the prevalence reported in Ghana (46%) but is lower than in Gabon (65%) [35,36]. Differences in parasite transmission, environmental factors, socioeconomic status, detection methods, and the repertoire of parasites detected likely account for these variations.

## Intestinal parasite infections

In this study, 43.1% of the pregnant women were infected with intestinal parasites, which was comparable with reports from western Ethiopia (43.8%) and northeast Ethiopia (43.5%) [37,38]. The finding was relatively higher than similar studies in India (8.0%) [39] and in northwest Ethiopia (37.3%) [26], but lower than studies in northwest Ethiopia (70.6%) [15] and Venezuela (73.9%) [40]. These variations might be due to the differences in parasite prevalence in the study areas, environmental factors, the accessibility of hygiene and sanitary facilities in the study areas, as well as the intervention measures taken.

This study identified hookworm as the leading cause of intestinal parasite infection (20.4%), followed by *S. mansoni* (12.5%) and *G. lamblia* (8%) among pregnant women. These infections might have a significant impact on pregnant women and their fetus since hookworm and schistosome infections are among the major causes of anemia in pregnancy, whereas giardiasis is a cause of malnutrition, and both anemia and malnutrition are predictors of adverse pregnancy outcomes [13]. Similarly, a recent meta-analysis study in Ethiopia reported that hookworm and *G. lamblia* were the most prevalent intestinal helminth and protozoan during pregnancy, respectively [8].

Intestinal helminths are soil-transmitted or water-associated, while protozoan infections are usually waterborne; hence, hygiene and sanitation are critical factors in the transmission of these infections. This study revealed significantly increased odds of intestinal parasitic infections among pregnant women who lived in rural areas. Moreover, intestinal parasitic

infections were significantly associated with a lack of practices of the recommended intestinal parasite prevention techniques, such as failure to use latrine consistently, drinking water from unprotected sources, and having risky habits such as consuming raw vegetables and soil (geophagia) during pregnancy. Similar findings were reported from different parts of Ethiopia [15,41,42]. This might be due to the fact that in the rural settings of Ethiopia, illiteracy level is high, the availability of toilet facilities is low, and access to clean water is low. Thus, in the rural areas, there is more open defecation, poor environmental sanitation, and poor personal hygiene, as well as lower access to health care facilities than the urban dwellers. On top of these, farming is the main economic activity in rural areas, which keep the rural population in consistent contact with infected soil and hence at increased risk of soil-transmitted helminth infections.

### *Plasmodium* infections

Among first ANC attending women, 19.1% of them were infected with *Plasmodium* species and *P. falciparum* was accounted for more than 80% of the infections, which was in line with a previous community-based study in the area [22]. Moreover, the magnitude of *Plasmodium* infection in this study was comparable with a similar study in Ghana (20.4%) [43]. However, it was lower than the reports from India (29.3%) [44], Benin (39.5%) and Burkina Faso (42%) [45], and higher than reports from Gambia (8.8%) [45]. The difference might be due to the variation in malaria transmission intensities among the study areas, the socio-economic and socio-demographic characteristics of the study populations, and variations in the diagnostic methods used.

The current study revealed a 17.3% prevalence of asymptomatic *Plasmodium* infections, which accounted for about 85% of the total *Plasmodium* infections. These asymptomatically infected women would be left undiagnosed and untreated, thus becoming a source of new malaria cases and a hub of transmission [12], since symptomatic pregnant women are usually diagnosed in Ethiopia [18]. Moreover, these women would suffer from the resulting adverse pregnancy outcomes, such as maternal anemia and low birth weight [12]. The finding was relatively higher than similar studies from Ghana (10.7%) [46] and Kenya (12.9%) [47], but lower than Tanzania (36.4%) [48]. The differences might be due to variations in malaria transmission intensities in the areas, the efficiency of the diagnostic methods used, and the socio-demographic and clinical profile of the study populations considered.

Moreover, the current study revealed that about 40% of *Plasmodium*-infected women had submicroscopic infections and more than 24% had subpatent infections, indicating that a substantial number of cases were left undetected by the standard diagnostic methods utilized in the study area. These undetected infections could be a potential source of new malaria cases in the community and might cause adverse pregnancy outcomes [12]. Therefore, asymptomatic, submicroscopic, and subpatent *Plasmodium* infections could pose a threat to infected women, the community, and the ongoing national malaria elimination efforts.

In this study, adolescent age and primigravidity were among the risk factors for *Plasmodium* infections. During malaria in pregnancy, the unique variant surface antigen 2-chondroitin sulfate A protein (VAR2CSA) is expressed on *P. falciparum*-infected erythrocytes and mediates placental sequestration by binding to chondroitin sulfate A expressed on syncytiotrophoblasts that line the placental intervillous spaces [49]. Thus, pregnant women develop gravidity and exposure-dependent antimalarial immunity, which might result in lower susceptibility and severity to upcoming *Plasmodium* infections [50]. Primigravid women lack previous exposure to VAR2CSA and thus might lack immunity to infected erythrocytes expressing this protein. In this study, adolescent and primigravid pregnant

women had more than twice increased risk of *Plasmodium* infections compared to their adult and multigravid counterparts. This was reported in previous studies [48,51,52].

The practice of vector control measures like consistent and proper use of ITN is among the pillars of malaria prevention [18]. This study showed a low proportion of ITN usage (36.8%), and pregnant women who did not consistently slept under ITN had more than double the risk of *Plasmodium* infections compared to those who sleep under an ITN consistently. In line with our findings, low usage of ITN among Ethiopian pregnant women was reported [53]. The inverse relationship between ITN use and *Plasmodium* infection in pregnancy was also widely reported [27,36,54].

The current study revealed that undernourished and intestinal helminth-infected ANC1-ttending pregnant women had about twice the increased risk of *Plasmodium* infections than well-nourished and intestinal helminth-non-infected pregnant women. This might be due to the fact that undernutrition may worsen the susceptibility of pregnant women to *Plasmodium* infection and could impair the development of protective immunity to malaria [55]. Similarly, intestinal helminths have an immunomodulatory effect, which is characterized by an increased level of anti-inflammatory cytokines, which could make helminth-infected women more susceptible to *Plasmodium* infections, and the increased mosquito-attractive nature of helminth-infected hosts due to helminth-induced anemia [56,57]. A study in Ghana showed that helminth-infected pregnant women had a more than four-fold increased risk of *Plasmodium* infection than helminth-non-infected women [58].

### *Plasmodium* and intestinal parasite coinfections

*Plasmodium* and intestinal parasite infections have different pathophysiologic mechanisms to induce anemia and adverse pregnancy outcomes; hence, their effects are synergized during coinfection [9]. During pregnancy, *P. falciparum*-infected erythrocytes sequester in the placenta and affect placental vasculogenesis, angiogenesis, and nutrient transport, which cause adverse pregnancy outcomes [59]. *Plasmodium*-induced hemolysis of the red blood cells, dysregulation and/or suppression of erythropoiesis, and destruction of non-parasitized red blood cells are the mechanisms by which malaria causes anemia [60]. On the other hand, intestinal parasite infections cause reduced food intake, malabsorption, and endogenous nutrient loss, which results in maternal anemia and malnutrition and thus adverse pregnancy outcomes [61]. In this study, coinfection with intestinal parasites and *Plasmodium* species occurred in roughly 12% of the pregnant women.

*Plasmodium*-intestinal parasite coinfection was independently predicted by rural residency, undernutrition, and primigravidity. Rural women may have a greater prevalence of coinfection because they are less likely to adopt the recommended preventative and control measures for intestinal parasites and malaria because of their higher degree of illiteracy. Due to parasite-induced decreased food intake, malabsorption, and endogenous nutritional loss, intestinal parasite infections may be linked to undernutrition [13]. On the other hand, undernourished women may be more susceptible to contracting *Plasmodium* infections and have weakened malaria protective immunity [55], leading to a higher incidence of *Plasmodium*-intestinal parasite coinfection. Moreover, lack of gravidity and exposure-dependent antimalarial immunity among primigravid women [50] might be the possible reason for the increased occurrence of *Plasmodium* infections among them compared to multigravidas. Similar findings were reported from other malaria-endemic countries in Africa [35,58].

Altogether, the current study provided data on *Plasmodium* and intestinal parasite infections and their determinants among pregnant women at ANC1, which was lacking in Ethiopia. However, the findings of the study should be considered alongside its limitations. The molecular testing of *Plasmodium* infection was conducted for only 78% of participants.

Moreover, the study did not include participants from the community other than ANC1-attending women, which could be helpful to evaluate the degree of community-level transmission reflected by ANC1-attending women.

## Conclusion

Overall, a high prevalence of delayed ANC1, *Plasmodium* infections, and intestinal parasite infections were observed among pregnant women in Jawi District, northwest Ethiopia. Significantly increased risk of *Plasmodium* and intestinal parasite infection was observed among younger, primigravid and rural resident women. Moreover, *Plasmodium* infection was largely asymptomatic, and a substantial proportion was submicroscopic, indicating the majority of *Plasmodium* infections would go undiagnosed and untreated. These might cause adverse pregnancy outcomes and a reservoir of malaria transmission in the community, thus hindering the ongoing malaria elimination efforts.

Therefore, strengthening prevention and control measures for parasitic infections, promoting early initiation of ANC follow-up, and implementing malaria screening at ANC1, particularly for young and primigravid women, are recommended to alleviate the burden. Further studies evaluating the effectiveness of the prevention and control measures for parasitic infections in pregnancy are suggested for timely interventions.

## Supporting information

**S1 Table. Primers and probes used for detection of *Plasmodium* species in pregnancy, northwest Ethiopia, 2022.**
(DOCX)

**S2 Data. Raw data collected from the study participant first antenatal care attending pregnant women, northwest Ethiopia, 2022.**
(XLSX)

## Acknowledgement

We are indebted to pregnant women for their participation in the study, the officers of the Jawi District Health Office and the medical directors of Jawi Primary Hospital, Jawi Health Center, and Bambluk Health Center for their cooperation. We would like to thank medical laboratory and midwifery professionals working in Jawi Primary Hospital, Jawi Health Center, and Bambluk Health Center for their technical assistance in the laboratory and antenatal care rooms.

Moreover, we are grateful to the Ethiopian Public Health Institute for allowing us to use their National Parasitology Reference Laboratory to perform the qPCR assay.

## Author contributions

**Conceptualization:** Zemenu Tamir, Abebe Animut, Sisay Dugassa, Aster Tsegaye, Berhanu Erko.

**Data curation:** Zemenu Tamir, Adugna Abera.

**Formal analysis:** Zemenu Tamir, Mahlet Belachew, Adugna Abera, Aster Tsegaye.

**Funding acquisition:** Abebe Animut, Berhanu Erko.

**Investigation:** Zemenu Tamir, Mahlet Belachew, Adugna Abera.

**Methodology:** Zemenu Tamir, Abebe Animut, Sisay Dugassa, Mahlet Belachew, Adugna Abera, Aster Tsegaye, Berhanu Erko.

**Resources:** Abebe Animut, Sisay Dugassa, Adugna Abera, Aster Tsegaye, Berhanu Erko.

**Software:** Zemenu Tamir.

**Supervision:** Berhanu Erko.

**Writing – original draft:** Zemenu Tamir.

**Writing – review & editing:** Zemenu Tamir, Abebe Animut, Sisay Dugassa, Mahlet Belachew, Adugna Abera, Aster Tsegaye, Berhanu Erko.

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
