## [Decision Letter · Decision Letter 0]

19 Feb 2024

PONE-D-23-31913Plasmodium and intestinal parasite infections among pregnant women at first antenatal care contact in northwest Ethiopia: a study of prevalence and associated risk factorsPLOS ONE

Dear Dr. Tamir,

Thank you for submitting your manuscript to PLOS ONE. After careful consideration, we feel that it has merit but does not fully meet PLOS ONE’s publication criteria as it currently stands. Therefore, we invite you to submit a revised version of the manuscript that addresses the points raised during the review process.

Before considering the manuscript for publication, address all the comment given by both reviewers.** ** In addition to reviewers' comment, include operational for 'sub-microscopic malaria'. 

We look forward to receiving your revised manuscript.

Kind regards,

Musa Mohammed Ali, PhD

Academic Editor

PLOS ONE

Journal Requirements:

Reviewers' comments:

Reviewer's Responses to Questions

**Comments to the Author**

1. Is the manuscript technically sound, and do the data support the conclusions?

Reviewer #1: Yes

Reviewer #2: Yes

2. Has the statistical analysis been performed appropriately and rigorously? 

Reviewer #1: No

Reviewer #2: Yes

3. Have the authors made all data underlying the findings in their manuscript fully available?

Reviewer #1: Yes

Reviewer #2: Yes

4. Is the manuscript presented in an intelligible fashion and written in standard English?

Reviewer #1: Yes

Reviewer #2: Yes

5. Review Comments to the Author

Reviewer #1: Line 116: Study design and population: The study population is not captured under this section. Title the section as: Study design and participants.

Line 118: The study arm is already captured in line 92-93 and should be delated from line 118.

Line 279-283: What informed the use of P<0.25?

The authors indicated in Line 279-283 that they are using P<0.025 but in Table 5 they are using 95%CI. Why not using P<0.05 that goes with 95%CI?

Tables 5, 6 and 7: Authors should be consistent with the position of reference category. The reference category is normally the category on the first row.

Lines 273-275: “The prevalence of intestinal parasites and Plasmodium species infections was estimated by dividing the number of pregnant women diagnosed positive by the total number of pregnant women examined.”

The definition of N in the line 273-275 is different from what is in Tables 5, 6 and 7.

What are the authors using total number of women in that category as denominator instead of using the the total number of pregnant women examined? These inconsistencies must be resolved with.

What are some of the limitations of the study?

Reviewer #2: 1. Better to include your study design in the method part of your abstract.

2. Include the result of some of your associated factors in the result section of the abstract.

3. How you calculated the parasitemia level for Plasmodium infection among the pregnant women? I think better to include.

4. What criteria is used to classify/categorize age?

Overall, the manuscript is well written and organized except the above few comments/suggestions.

6. PLOS authors have the option to publish the peer review history of their article (what does this mean? ). If published, this will include your full peer review and any attached files.

**Do you want your identity to be public for this peer review?** For information about this choice, including consent withdrawal, please see our Privacy Policy .

Reviewer #1: No

Reviewer #2: **Yes: ** Tadesse Duguma

---

## [Author Response · Author response to Decision Letter 1]

25 Feb 2024

Dear Sir/Madam,

We would like to thank you and the reviewers for the constructive comments made to bring our paper to the required standard. We have also addressed all the comments made by you and the reviewers in the revised version of the paper and provided point-by-point rebuttal to each comment and suggestion made. The responses to specific Editor's and reviewers' comments are addressed and uploaded in the 'Response to Reviewers' letter.

---

## [Decision Letter · Decision Letter 1]

23 Jun 2024

PONE-D-23-31913R1Plasmodium and intestinal parasite infections among pregnant women at first antenatal care contact in northwest Ethiopia: a study of prevalence and associated risk factorsPLOS ONE

Dear Dr. Tamir,

Thank you for submitting your manuscript to PLOS ONE. After careful consideration, we feel that it has merit but does not fully meet PLOS ONE’s publication criteria as it currently stands. Therefore, we invite you to submit a revised version of the manuscript that addresses the points raised during the review process.

We look forward to receiving your revised manuscript.

Kind regards,

Musa Mohammed Ali, PhD

Academic Editor

PLOS ONE

Reviewers' comments:

Reviewer's Responses to Questions

**Comments to the Author**

1. If the authors have adequately addressed your comments raised in a previous round of review and you feel that this manuscript is now acceptable for publication, you may indicate that here to bypass the “Comments to the Author” section, enter your conflict of interest statement in the “Confidential to Editor” section, and submit your "Accept" recommendation.

Reviewer #2: (No Response)

Reviewer #3: (No Response)

2. Is the manuscript technically sound, and do the data support the conclusions?

Reviewer #2: Yes

Reviewer #3: No

3. Has the statistical analysis been performed appropriately and rigorously? 

Reviewer #2: Yes

Reviewer #3: Yes

4. Have the authors made all data underlying the findings in their manuscript fully available?

Reviewer #2: Yes

Reviewer #3: Yes

5. Is the manuscript presented in an intelligible fashion and written in standard English?

Reviewer #2: No

Reviewer #3: No

6. Review Comments to the Author

Reviewer #2: My comments and suggestions

After looking in to the details of the manuscript, I do have few comments & suggestions indicated below:

1. Correct the scientific naming of malaria causing protozoan in some sections

(Table 3 first line, Table 5: at bottom line outside the table & Figure 3: caption).

2. Line 113 “Study participant women were recruited between November 27, 2021, and July 15, 2022” better to written as “Women study participants were recruited between November 27, 2021, and July 15, 2022”.

3. Need some sort of English language improvement

4. The first revision made to the manuscript fixed many of the concerns and now the manuscript is in a very good position to be published as per my view.

Overall, the manuscript is well written piece of scientific finding with only the few suggestions raised above to be considered before the article is accepted for publication.

Reviewer #3: Comments to the authors: Background

o The opening sentence is clear but can be made more concise and impactful.

o The prevalence data provided for malaria and intestinal parasites is informative but should be directly linked to the significance of the study.

o Clearly differentiate between the impacts of Plasmodium infections and intestinal parasite infections.

o Clearly outline the WHO recommendations and Ethiopia’s efforts to reduce the burden of these infections.

o Clearly state the significance of the pooled prevalence data for malaria and intestinal parasite infections among pregnant women in Ethiopia.

o Emphasize the need for the study and the gap it aims to fill.

o Some statements are too broad or lack specific details.

o Provide more specific information and context for statements. For example, specify the types of interventions or public health efforts being referenced.

o The background does not sufficiently address the role of socio-economic factors in the prevalence of malaria and intestinal parasites.

o Include a discussion on how socio-economic factors, such as education, income, and access to healthcare, influence the prevalence and management of these infections.

o Some prevalence data and references might be outdated.

o Ensure that all data and references are up-to-date, reflecting the most recent research and statistics available.

o Limited comparison with global data outside the African region.

o Include comparisons with prevalence and risk factors data from other regions (e.g., Asia, South America) to provide a broader context and relevance of the study findings.

Methods

o Provide a clear description of the study area and its significance for the study.

o Clearly describe the study design and participant recruitment criteria.

o Provide a detailed and clear description of the data collection procedure.

o

o Provide a clear and detailed description of the laboratory procedures for diagnosing intestinal parasites and malaria.

o Some sentences are overly complex and lengthy, making them difficult to follow.

o Simplify sentences and use bullet points or lists for presenting complex data where appropriate.

Examples:

o "Among the diagnosed pregnant women, 272 (50.6%, 95% CI: 46.2–54.9) women were infected with at least a single species of Plasmodium or intestinal parasites."

o Revised: "Among the pregnant women, 50.6% (95% CI: 46.2–54.9) were infected with at least one species of Plasmodium or intestinal parasites."

o "Plasmodium and intestinal parasite co-infections were observed among 11.7% (63/538) of the pregnant women."

o Revised: "Co-infections with Plasmodium and intestinal parasites were found in 11.7% of the pregnant women."

o Data presentation can be confusing without proper context and clear formatting.

o Use tables and figures effectively, ensuring they are referenced in the text and properly labeled. Provide clear explanations for all data presented.

o Inconsistent use of terminology and statistical measures can lead to confusion.

o Ensure consistent use of terms (e.g., "Plasmodium infection" vs. "malaria infection") and statistical measures (e.g., OR, AOR, CI) throughout the section.

o References to unpublished documents (e.g., Jawi District Administrative Office) need to be substantiated with publicly available data or citations.

o Where possible, replace unpublished data with references to publicly available sources or provide a detailed description of the source in the manuscript.

o Some descriptions of laboratory techniques are too detailed and could be summarized.

o Provide essential details and refer to established protocols or previous studies for more comprehensive descriptions.

o The justification for the chosen sample size and sampling technique is not fully clear.

o Include a brief justification for the sample size determination and sampling technique used, explaining why they are appropriate for the study objectives.

Results

o The initial overview of participant characteristics should be concise and clear.

o Ensure the tables are clearly labeled and easy to interpret. Add titles and footnotes if necessary for better understanding.

o Provide a clear and concise summary of the prevalence findings.

2. Figures 3 and 4:Ensure figures are referenced correctly in the text and that legends clearly describe what the figures show.

o Suggested revision: "Figure 3 shows the distribution of Plasmodium species, the proportion of symptomatic and asymptomatic malaria cases, and the proportion of microscopic, submicroscopic, and subpatent Plasmodium infections. Figure 4 displays the overlap of Plasmodium species, intestinal helminth, and intestinal protozoa infections among the participants."

o Clearly present the factors associated with infections and their statistical significance.

3. Tables 5 and 6:

o Ensure tables are easy to read and understand, with clear headings and footnotes explaining abbreviations and statistical terms.

o No change to content but ensure tables are properly formatted and clearly labeled.

4. Co-Infections:

o Summarize the findings on co-infections clearly and concisely.

o Suggested revision: "Plasmodium and intestinal parasite co-infections were found in 11.7% of the pregnant women. Co-infections of Plasmodium with intestinal helminths, intestinal protozoa, and both were observed in 8.7%, 1.1%, and 1.9% of the women, respectively."

Discussion

o Start with a strong statement about the significance of the study findings.

2. Delayed ANC initiation:

o Clearly state the findings and their implications.

3. Prevalence and associated risk factors:

o Clearly present the prevalence and associated risk factors for infections.

o Compare findings with those from other studies clearly and concisely.

4. Hookworm and other intestinal parasites:

o Summarize the findings related to intestinal parasites and their implications.

o Discuss the role of hygiene and sanitation in infection transmission.

5. Malaria prevalence and asymptomatic infections:

o Highlight the significance of malaria prevalence and asymptomatic infections.

6. Submicroscopic and subpatent infections:

o Discuss the importance of detecting submicroscopic and subpatent infections.

7. Risk factors for malaria:

o Summarize the risk factors for malaria and their implications.

8. Co-infections:

o Discuss the impact of co-infections and associated risk factors.

9. Limitations:

o Acknowledge the study's limitations clearly and concisely.

o Some points are repeated multiple times, making the discussion lengthy and redundant.

o Combine repetitive points and ensure each paragraph adds new information or insights.

Example 1:

Original text:

1. "In the current study, about four out of five women had delayed ANC initiation, and one out of two women was infected with malaria or intestinal parasites. About 43% of women had intestinal parasite infections, 19% had malaria, and 12% were co-infected with malaria and intestinal parasites at ANC1."

2. "The high proportion of delayed ANC1 in this study was in line with the report of the National Demographic Health Survey [30]. Early initiation of ANC follow-up is highly recommended by the WHO for full benefit from the health promotion, disease prevention, and curative services provided on the platform [10]. In Ethiopia, particular interventions aimed for healthy pregnancy, such as deworming, the provision of ITN, and supplements (iron and folic acid), are provided on the ANC platform [13]. Thus, due to the late ANC1, the majority of pregnant women in the study area may not fully benefit from the services provided on the platform."

Revised text: "In the current study, approximately 80% of women had delayed ANC initiation, and 50% were infected with malaria or intestinal parasites at ANC1. Specifically, 43% had intestinal parasite infections, 19% had malaria, and 12% were co-infected. This high rate of delayed ANC initiation aligns with the National Demographic Health Survey [30]. WHO recommends early ANC initiation for optimal health promotion, disease prevention, and curative services [10]. In Ethiopia, ANC services include deworming, ITN provision, and supplements (iron and folic acid) [13]. Consequently, delayed ANC1 means many pregnant women may not fully benefit from these services."

Example 2:

Original text:

1. "It has been indicated that pregnant women are more susceptible to infections than non-pregnant women [2]. Moreover, the prevalence of these infections at ANC1 is a reflection of the transmission level and intervention coverage in the community [18]."

2. "In the current study, more than half of the pregnant women were infected with Plasmodium and/or intestinal parasites at ANC1, revealing a high burden of parasites in the study area, which was supported by previous reports [20, 31]. The current finding was higher than a report from Ghana (46%) [32] but lower than that from Gabon (65%) [33]. The difference might be due to the parasite transmission level of the study areas during the study period, climatic and environmental factors, the socioeconomic status of the population, the methods used for parasite detection, and the repertoire of parasites detected in the studies."

Revised text: "Pregnant women are more susceptible to infections than non-pregnant women [2], and the prevalence of infections at ANC1 reflects community transmission levels and intervention coverage [18]. In this study, over half of the pregnant women were infected with Plasmodium and/or intestinal parasites at ANC1, indicating a high burden of parasites, consistent with previous reports [20, 31]. This finding exceeds the prevalence reported in Ghana (46%) but is lower than in Gabon (65%) [32, 33]. Differences in parasite transmission, environmental factors, socioeconomic status, detection methods, and the repertoire of parasites detected likely account for these variations."

o Some sentences are overly complex and lengthy, making them difficult to follow.

o Simplify sentences and ensure each one conveys a single clear idea.

Examples:

o Original: "The high proportion of delayed ANC1 in this study was in line with the report of the National Demographic Health Survey [30]. Early initiation of ANC follow-up is highly recommended by the WHO for full benefit from the health promotion, disease prevention, and curative services provided on the platform [10]."

o Revised: "The high proportion of delayed ANC1 in this study aligns with the National Demographic Health Survey [30]. WHO recommends early ANC initiation for optimal health promotion, disease prevention, and curative services [10]."

o Some findings are not clearly presented or are buried in complex sentences.

o Use clear and straightforward language to present key findings and their implications.

Conclusion

o Begin with a clear and concise summary of the overall findings.

o Summarize specific key findings without repeating detailed statistics.

o Clearly state the implications of these findings.

o End with a clear call to action or recommendations based on the findings.

7. PLOS authors have the option to publish the peer review history of their article (what does this mean? ). If published, this will include your full peer review and any attached files.

**Do you want your identity to be public for this peer review?** For information about this choice, including consent withdrawal, please see our Privacy Policy .

Reviewer #2: **Yes: ** Tadesse Duguma

Reviewer #3: No

---

## [Author Response · Author response to Decision Letter 2]

17 Aug 2024

Point-by-Point Responses to the Editor and Reviewers Comments

Dear Sir/Madam,

We would like to thank you and the reviewers for the constructive comments made to bring our paper to the required standard. We have also addressed all the comments made by the editor and the reviewers in the revised version of the paper and provided point-by-point rebuttal to each comment and uploaded as "Response to reviewers' file.

---

## [Decision Letter · Decision Letter 2]

22 Sep 2024

PONE-D-23-31913R2Plasmodium and intestinal parasite infections among pregnant women at first antenatal care contact in northwest Ethiopia: a study of prevalence and associated risk factorsPLOS ONE

Dear Dr. Tamir,

Thank you for submitting your manuscript to PLOS ONE. After careful consideration, we feel that it has merit but does not fully meet PLOS ONE’s publication criteria as it currently stands. Therefore, we invite you to submit a revised version of the manuscript that addresses the points raised during the review process.

We look forward to receiving your revised manuscript.

Kind regards,

Musa Mohammed Ali, PhD

Academic Editor

PLOS ONE

Reviewers' comments:

Reviewer's Responses to Questions

**Comments to the Author**

1. If the authors have adequately addressed your comments raised in a previous round of review and you feel that this manuscript is now acceptable for publication, you may indicate that here to bypass the “Comments to the Author” section, enter your conflict of interest statement in the “Confidential to Editor” section, and submit your "Accept" recommendation.

Reviewer #2: (No Response)

Reviewer #3: All comments have been addressed

2. Is the manuscript technically sound, and do the data support the conclusions?

Reviewer #2: Yes

Reviewer #3: Yes

3. Has the statistical analysis been performed appropriately and rigorously? 

Reviewer #2: Yes

Reviewer #3: Yes

4. Have the authors made all data underlying the findings in their manuscript fully available?

Reviewer #2: Yes

Reviewer #3: Yes

5. Is the manuscript presented in an intelligible fashion and written in standard English?

Reviewer #2: Yes

Reviewer #3: Yes

6. Review Comments to the Author

Reviewer #2: My comments regarding the revised manuscript titled as “Plasmodium and intestinal parasite infections among pregnant women at first antenatal care contact in northwest Ethiopia: a study of prevalence and associated risk factors”

First, I need to forward my appreciation to the authors of the manuscript for the comprehensive work!

What is your reference to say a high prevalence of malaria and intestinal parasites detected in conclusion part of your abstract (line 44)? What is your base line?

Could you describe the magnitude of gametocyte carriage among the pregnant women which is known to play an important role in maintain malaria transmission in a community?

Line 289-290: Only variables with P-values <0.05 in the bivariable logistic regression analysis were included in the multivariable logistic regression model? I think variables with P-value of < 0.25 in bivariable analysis are candidates for multivariable logistic regression.

What is your reference for age category used in this manuscript for pregnant women? Even it is not consistent throughout the document Table 1 vs Table (5,6,7)?

Overall, the manuscript is interesting and well written except the comments and suggestions raised above.

Reviewer #3: Review of the Introduction:

• Overall, the grammar is acceptable, but there are areas where sentence structure can be tightened to enhance clarity and readability. Some sentences feel unnecessarily long and can be simplified without losing meaning.

o "More than 13 million pregnant women were exposed to malaria in 2021 in 38 malaria-endemic countries in the World Health Organization’s (WHO) African Region."

Could be: "In 2021, more than 13 million pregnant women in 38 malaria-endemic countries within the World Health Organization’s (WHO) African Region were exposed to malaria."

o "However, malaria and intestinal parasite infections still remain major public health problems in the country, particularly among at-risk population groups like pregnant women."

Could be: "Despite efforts, malaria and intestinal parasite infections continue to be major public health issues in Ethiopia, especially among vulnerable populations such as pregnant women."

o "Intestinal parasitosis causes anorexia, abdominal pain, nausea, vomiting, diarrhea, and intestinal bleeding."

Consider merging the lists or varying sentence length for better readability: "Intestinal parasitosis results in symptoms such as anorexia, abdominal pain, nausea, vomiting, diarrhea, and even intestinal bleeding."

• The flow between the general context of malaria/intestinal parasitic infections and the discussion of risk factors is a bit abrupt. More could be done to introduce the section on risk factors and interventions.

Adding a transition sentence after discussing the health impact of these infections could improve the flow. For example:

o "Given the significant health implications of Plasmodium and intestinal parasite infections, it is crucial to understand the underlying risk factors that exacerbate these conditions."

While the introduction is generally well-written, the tone sometimes lacks precision in linking global data with local data from Ethiopia.

Strengthen the connection between global figures and local context by specifying how Ethiopia fits into the broader landscape. For example, after discussing the global burden of malaria among pregnant women, directly segue into Ethiopian figures:

There is an inconsistency in the use of both "Plasmodium infections" and "malaria." While "Plasmodium" is correct scientifically, alternating between these terms might cause confusion for the reader.

Stick to one terminology. For example, "malaria" can be used when discussing clinical aspects, and "Plasmodium infections" when focusing on parasitology.

• Some sentences are wordy, making them harder to read quickly.

o Example: "Pregnant women are more susceptible to infections as a result of pregnancy-associated changes."

Could be: "Pregnancy-associated changes increase susceptibility to infections in women."

• The paragraph discussing maternal age and socio-economic factors as predictors of infection is dense. This information could be better organized with a clearer separation between biological risk factors and socio-economic ones. Breaking it into two smaller paragraphs would improve readability.

• Introduce abbreviations such as ANC (antenatal care) earlier in the introduction to avoid repetition of the full term.

Methods Section

The cross-sectional study design is appropriate given the objective of determining the prevalence and risk factors of Plasmodium and intestinal parasite infections. However, a stronger justification for using this design could be added. The current phrasing simply states that it is a "health-facility-based cross-sectional study." Consider adding a sentence that clarifies why this design is best suited for your research goals. For example:

"This design was chosen as it allows for the rapid collection of prevalence data at a single point in time, ideal for assessing disease burden in specific populations."

• Many sentences in the methods section could benefit from simplification to improve readability. For example:

o "Pregnant women who met the eligibility criteria and consented to participate were included in the study."

• The explanation of the sample size formula is clear, but it would be beneficial to provide a brief explanation of why you used 8.2% and 37.3% as the expected prevalences for Plasmodium and intestinal parasites. Adding references or justification from previous studies in similar regions would strengthen this section.

• The section on data collection is well detailed but could benefit from a clearer breakdown. For example, explaining the sequence of steps for the stool and blood sample collection could be clarified.

o Could be: "Data collection involved two steps: first, stool samples were collected and examined microscopically using the wet mount and Kato-Katz techniques for intestinal parasites. Blood samples were then collected for Plasmodium diagnosis via light microscopy, rapid diagnostic tests (RDT), and real-time PCR."

• The transition between study design, sample size determination, and data collection is smooth, but there is a slight disconnect when introducing statistical analysis. It may help to mention the types of variables being analyzed before introducing logistic regression.

After discussing data collection, a sentence introducing the analysis variables can improve flow. For instance: "The collected data included demographic, environmental, and behavioral variables, which were used in subsequent statistical analyses."

The use of logistic regression is appropriate, but there’s a lack of clarity on how risk factors were selected. Were they chosen based on prior knowledge or identified during the study? Clarifying this will align better with the study objectives.

• The phrase "intestinal parasitosis" is used interchangeably with "intestinal parasite infections." While both terms are correct, it is better to stick to one term throughout the section to avoid confusion.

Use "intestinal parasite infections" consistently, as it aligns more closely with the title.

• The methods for diagnosing malaria and intestinal parasites (light microscopy, RDT, PCR) are comprehensive. However, more detail on the reliability and justification for using multiple diagnostic tools (especially for malaria) would enhance the section. For example, why was real-time PCR necessary alongside RDT?

o Could be: "While RDT and microscopy are widely used for malaria diagnosis, real-time PCR was employed to detect sub-microscopic infections, ensuring a more accurate estimation of Plasmodium prevalence."

• The ethical section is concise, but adding more detail about how participant confidentiality was maintained could strengthen it.

Results Section

3. Grammar and Style:

• Many sentences are overly complex, making them harder to read. Simplifying sentence structure would improve readability.

o Example:

Original: "Among the pregnant women, 50.6% (95% CI: 46.2–54.9) were infected with at least a single species of Plasmodium or intestinal parasites."

Could be: "In total, 50.6% (95% CI: 46.2–54.9) of the pregnant women had either Plasmodium or intestinal parasite infections."

• There are a few instances where the passive voice obscures clarity. Active voice should be used more often to clarify who performed actions.

o Example:

Original: "Blood samples were analyzed by light microscopy and PCR."

Could be: "We analyzed blood samples using light microscopy and PCR."

• The use of "intestinal parasitosis" and "intestinal parasite infections" is inconsistent. Stick to one term throughout the section. Since "intestinal parasite infections" is used more often, it is advisable to stick with that term for clarity.

• The prevalence and confidence intervals are appropriately reported. However, there is room for improvement in interpretation, particularly in clarifying the significance of subgroups such as microscopic, sub-microscopic, and subpatent Plasmodium infections.

o Explain briefly what "sub-microscopic" and "subpatent" infections mean and their significance in the context of public health.7

• Ensure acronyms are introduced the first time they appear (e.g., RDT is introduced without being spelled out).

• The discussion effectively addresses the study's objectives, linking the findings back to the prevalence and risk factors of Plasmodium and intestinal parasite infections. However, the focus on first antenatal care (ANC1) should be highlighted more consistently. In some parts of the discussion, ANC1 is implied but not explicitly mentioned.

• The section flows logically from discussing delayed ANC initiation, to prevalence rates, and finally to risk factors and comparisons with other studies. However, it would benefit from a clearer structure, particularly in separating findings on Plasmodium from intestinal parasites and co-infections.

o Consider dividing the discussion into subsections focusing on (1) Plasmodium infections, (2) Intestinal parasitic infections, and (3) Co-infections. This will help in maintaining focus and improving the readability of the discussion.

• The comparison with Ghana and Gabon is appropriate, but the rationale behind these geographical comparisons could be more robust. Provide a stronger explanation for why these countries were chosen for comparison, such as similar endemic conditions or socio-economic factors.

• While passive voice is common in scientific writing, excessive use can make the text less engaging. For instance:

"Approximately 80% of women had delayed ANC initiation."

Could be: "In our study, 80% of women delayed initiating ANC."

• Some phrases can be simplified for better readability.

o Example: "Approximately four out of five women had delayed ANC initiation."

Could be: "Eighty percent of the women delayed ANC initiation."

• The discussion of risk factors such as rural residency, lack of latrines, and soil consumption is appropriate. However, more interpretation could be added regarding why these factors specifically contribute to higher infection rates in this population. For example, explaining how rural residency may relate to access to healthcare or preventive measures like insecticide-treated nets would strengthen the discussion.

• The discussion on co-infections could be expanded to emphasize the synergistic effects of Plasmodium and intestinal parasites on pregnancy outcomes. For instance, how these co-infections might exacerbate maternal anemia or increase the risk of adverse birth outcomes.

• Throughout the discussion, there is some inconsistency in referring to infections as "intestinal parasitosis" and "intestinal parasite infections." Choose one term to use consistently throughout the manuscript. "Intestinal parasite infections" is more straightforward and widely understood in the context of public health.

• The citations of other studies, such as comparisons with Ghana and Gabon, are useful. However, some claims are made without immediate reference to supporting literature. Ensure that every claim, especially when referring to public health recommendations or the impacts of infections, is properly cited.

• The limitations of the study are appropriately acknowledged, including the lack of molecular testing for all participants and the restriction of the study to ANC1-attending women. However, there is room to strengthen this section by discussing potential confounding factors or selection bias, particularly since the study focuses only on women attending healthcare facilities, which may not reflect the broader population.

• Ensure consistency when reporting statistical findings. In the discussion, terms like "odds ratios" and "adjusted odds ratios" should be consistently abbreviated as OR or AOR after the first mention.

Conclusion Section

The conclusion effectively summarizes the high prevalence of both malaria and intestinal parasitic infections among pregnant women, particularly in first antenatal care visits (ANC1). This aligns well with the study's objective to assess prevalence and risk factors.

• Some parts of the conclusion can be phrased more concisely. For example:

o Original: "Thus, strengthening of parasitic infection prevention and control measures and screening of malaria at ANC1, particularly in young and primigravid women, are suggested."

o Could be: "Strengthening prevention and control measures for parasitic infections and implementing malaria screening at ANC1, particularly for young and primigravid women, are recommended."

• Use more active voice to make the conclusion more engaging.

o Example: "These findings highlight the importance of..." instead of "Thus, it is suggested..."

• smoothly from discussing prevalence to recommending interventions. However, it could include a more comprehensive summary of the study’s findings to reinforce their significance. For example:

Briefly restating key findings on the association between maternal age, rural residency, and primigravidity with higher infection rates can help the reader remember these critical points.

• The recommendations made are appropriate, but the suggestions for early ANC initiation could be emphasized further. Since one of the main findings was the high rate of delayed ANC initiation (80%), it would be effective to elaborate on how promoting timely ANC visits can help mitigate these infections.

• Consider adding a brief mention of the need for future research or longitudinal studies to monitor the effectiveness of the recommended interventions over time.

7. PLOS authors have the option to publish the peer review history of their article (what does this mean? ). If published, this will include your full peer review and any attached files.

**Do you want your identity to be public for this peer review?** For information about this choice, including consent withdrawal, please see our Privacy Policy .

Reviewer #2: **Yes: ** Tadesse Duguma Adula

Reviewer #3: No

---

## [Author Response · Author response to Decision Letter 3]

4 Oct 2024

Dear Sir/Madam,

We would like to thank the editor and the reviewers for the constructive comments made to bring our paper to the required standard. We have addressed all the journal requirements and comments made by the editor and the reviewers in the revised version of the paper and provided point-by-point rebuttals to each comment and suggestion and uploaded it in a file labeled "Response to reviewers" letter.

---

## [Editor Report · Decision Letter 3]

12 Dec 2024

Plasmodium and intestinal parasite infections among pregnant women at first antenatal care contact in northwest Ethiopia: a study of prevalence and associated risk factors

PONE-D-23-31913R3

Dear Dr. Tamir,

We’re pleased to inform you that your manuscript has been judged scientifically suitable for publication and will be formally accepted for publication once it meets all outstanding technical requirements.

Kind regards,

Luzia H Carvalho, Ph.D.

Academic Editor

PLOS ONE
---

## [Editor Report · Acceptance letter]

PONE-D-23-31913R3

PLOS ONE

Dear Dr. Tamir,

I'm pleased to inform you that your manuscript has been deemed suitable for publication in PLOS ONE. Congratulations! Your manuscript is now being handed over to our production team.

Kind regards,

on behalf of

Dr. Luzia H Carvalho

Academic Editor

PLOS ONE